# Mammographic Density and Screening Sensitivity, Breast Cancer Incidence and Associated Risk Factors in Danish Breast Cancer Screening

**DOI:** 10.3390/jcm8112021

**Published:** 2019-11-19

**Authors:** Elsebeth Lynge, Ilse Vejborg, Zorana Andersen, My von Euler-Chelpin, George Napolitano

**Affiliations:** 1Nykøbing Falster Hospital, University of Copenhagen, Ejegodvej 63, DK-4800 Nykøbing Falster, Denmark; 2Radiology Clinic, Copenhagen University Hospital, Rigshospitalet, DK-2100 København Ø, Denmark; ilse.vejborg@regionh.dk; 3Department of Public Health, University of Copenhagen, DK-1014 København K, Denmark; zorana.andersen@sund.ku.dk (Z.A.); myeu@sund.ku.dk (M.v.E.-C.); gena@sund.ku.dk (G.N.)

**Keywords:** mammographic density, sensitivity, breast cancer incidence, risk factor, body constitution

## Abstract

Background: Attention in the 2000s on the importance of mammographic density led us to study screening sensitivity, breast cancer incidence, and associations with risk factors by mammographic density in Danish breast cancer screening programs. Here, we summarise our approaches and findings. Methods: Dichotomized density codes: fatty, equal to BI-RADS density code 1 and part of 2, and other mixed/dense data from the 1990s—were available from two counties, and BI-RADS density codes from one region were available from 2012/13. Density data were linked with data on vital status, incident breast cancer, and potential risk factors. We calculated screening sensitivity by combining data on screen-detected and interval cancers. We used cohorts to study high density as a predictor of breast cancer risk; cross-sectional data to study the association between life style factors and density, adjusting for age and body mass index (BMI); and time trends to study the prevalence of high density across birth cohorts. Results: Sensitivity decreased with increasing density from 78% in women with BI-RADS 1 to 47% in those with BI-RADS 4. For women with mixed/dense compared with those with fatty breasts, the rate ratio of incident breast cancer was 2.45 (95% CI 2.14–2.81). The percentage of women with mixed/dense breasts decreased with age, but at a higher rate the later the women were born. Among users of postmenopausal hormone therapy, the percentage of women with mixed/dense breasts was higher than in non-users, but the patterns across birth cohorts were similar. The occurrence of mixed/dense breast at screening age decreased by a z-score unit of BMI at age 13—odds ratio (OR) 0.56 (95% CI 0.53–0.58)—and so did breast cancer risk and hazard ratio (HR) 0.92 (95% CI 0.84–1.00), but it changed to HR 1.01 (95% CI 0.93–1.11) when controlled for density. Age and BMI adjusted associations between life style factors and density were largely close to unity; physical activity OR 1.06 (95% CI 0.93–1.21); alcohol consumption OR 1.01 (95% CI 0.81–1.27); air pollution OR 0.96 (95% 0.93–1.01) per 20 μg/m^3^; and traffic noise OR 0.94 (95% CI 0.86–1.03) per 10 dB. Weak negative associations were seen for diabetes OR 0.61 (95% CI 0.40–0.92) and cigarette smoking OR 0.86 (95% CI 0.75–0.99), and a positive association was found with hormone therapy OR 1.24 (95% 1.14–1.35). Conclusion: Our data indicate that breast tissue in middle-aged women is highly dependent on childhood body constitution while adult life-style plays a modest role, underlying the need for a long-term perspective in primary prevention of breast cancer.

## 1. Introduction

In population-based breast cancer screening programs, all women in the target age-group are invited to screening, but not all of them attend. In these settings, breast cancer screening has been found to reduce breast cancer mortality by 20–25% for the entire population of women invited, and by 35% for women who attended and were actually screened [1]. However, this is an average, and the effect might vary across subgroups of women. From multiple publications from the beginning of the 2000s, it became clear that mammographic density was one of the most important factors associated with variations both in screening sensitivity [2], and in underlying risk of breast cancer [3]. 

Due to the use of personal identification numbers in nationwide population and health registers, Denmark offers exceptional possibilities for epidemiological research. Denmark, furthermore, established two pioneer breast cancer screening programs in the early 1990s and used a dichotomized code for mammographic density to schedule women for one or two-view mammography at the next screening. 

On this basis, using our population-based cohort, we first determined the association of mammographic density on screening sensitivity and breast cancer incidence, and second, searched for determinants of mammographic density with other data from these women. 

## 2. Material and Methods

### 2.1. Breast Cancer Screening and Density Coding in Denmark

In Denmark, breast cancer screening started in the municipality of Copenhagen in 1991 and in the county of Funen in 1993. Together, the two areas covered about 20% of the Danish population. Women aged 50–69 years were invited to screening every second year. At the first screening, all women had two-view mammography, which takes a craniocaudal and an oblique view [4,5]. All mammograms were read independently by two radiologists. Mammograms with no sign of malignancy, so-called negative mammograms, were coded as either fatty or mixed/dense breast tissue. Fatty was set to be equivalent to BI-RADS density code 1 and part of code 2 [4]. To save resources at subsequent screens, women with fatty breasts were offered only an oblique view at next screen, while women with mixed/dense breasts continued to be offered two-view mammography. This practice continued for the first ten years, where after all women were offered two-view mammography at each screen. In a small validation study, the dichotomous mammographic density code was found to agree well with the BI-RADS density codes [6]. As only negative mammograms were coded, no density code was available for breast cancers detected at the first screen, but all breast cancers detected at subsequent screens had a density code.

Breast cancer screening became nationwide in Denmark from 2007–2008, organized into five regional programs all using digital mammography. Since November 2012, mammograms in the Capital Region have been coded according to version 4 of the BI-RADS density code [7]. The density code was set independently by the two radiologists, and the highest score was used. The raw data from the digital mammograms have been stored for research on automated density and texture coding [8].

The mammography density data used for the studies reported in the following sections came from (1), the dichotomously coded mammograms from the two pioneer screening programs during 1991/93 to 2001; and from (2) the BI-RADS density coded mammograms from the Capital Region part of the national program during 2012–2013.

### 2.2. Other Data Sources

The cohorts of screened and density-coded women were followed up for death and emigration via linkage with the Central Population Register, and for incident cases of breast cancer via linkage with the Danish Cancer Register, Appendix A.

The dichotomously coded data from Copenhagen were further linked with the Copenhagen School Health Register, including weight and height measurements for women attending school in Copenhagen. Linkage was made also with the Copenhagen part of the Diet, Cancer and Health prospective cohort study. The dichotomously coded data from Funen were linked with the Odense University Pharmacoepidemiological Database [9]. The personal identification numbers were used for linkage.

### 2.3. Calculations

Four types of statistical analyses were made. First, sensitivity and specificity of screening by density was estimated from the screen-detected cancer cases and from the interval cancers. A screen-detected cancer was defined as an incident breast cancer diagnosed within 6 months of a positive screening mammogram. Interval cancers included two groups; first, a breast cancer diagnosed in a screen-negative woman within 2 years of the screen; second, a breast cancer diagnosed in a screen-positive woman 7 months to 2 years after a positive screen. Sensitivity was defined as (screen-detected cancers)/(screen-detected cancers and interval cancers). Specificity was defined as (screen-negative, breast cancer free women)/(breast cancer free women).

Second, breast cancer incidence by density was calculated for a long follow-up time based on the dichotomized data as the incident cases divided by the accumulated person years for screened women. The rate ratio of women with mixed/dense breasts compared with women with fatty breasts was calculated with Poisson regression controlling for age and calendar period. For a short follow-up time based on the BI-RADS density coded data, we calculated the risk of breast cancer for women with a given BI-RADS as the number of incident cases within 2 years divided by the number of women screened. Age-adjusted relative risks were calculated by using BI-RADS1 as baseline.

Third, the proportion of screened women with mixed/dense breasts was tabulated against variables of age, birth cohort, and use of postmenopausal hormone therapy. Fourth, cross-sectional associations between potential risk factors and the occurrence of mixed/dense breasts were estimated as odds ratios (OR) using multivariate logistic regression with controls for, among other factors, age and body mass index (BMI). Fifth, breast cancer incidence by use of postmenopausal hormone therapy or childhood body constitution combined with mammographic density at the first screen was calculated as the number of incident cases divided by the accumulated person years. Cox regression analysis was used, with age as the underlying time scale. The impact of childhood constitution was calculated as per z-score of BMI at a given age [10]. Various software, as specified in the original publications, were used for the analysis.

Approval of data handling by the Danish Data Protection Agency serves as ethical approval for register-based research in Denmark.

## 3. Results

### 3.1. Sensitivity/Specificity

During the first ten years of the Copenhagen pioneer program, 134,640 screening examinations were made, and 46% of them were coded as fatty (Table 1). The 61,741 screening examinations where women had fatty breasts included 159 women with screen-detected cancer (including carcinoma in situ), and 65 women with interval cancer. The 71,823 screening examinations where women had mixed/dense breasts included 312 screen-detected cancers (including carcinoma in situ), and 214 women with interval cancer. Screening sensitivity was higher for women with fatty breasts than for women with mixed/dense breasts; the age-adjusted odds ratio (OR) of interval cancer versus screen-detected cancer for women with mixed/dense breasts compared with women with fatty breasts was 1.62 (95% confidence interval (CI) 1.14–2.30) [4]. Screening sensitivity was, therefore, better for women with fatty breasts than for women with mixed/dense breasts, despite the fact that the first group had one-view mammography while the second group had two-view mammography.

In 2012/13, in total 54,808 women were screened in the Capital Region, and of these, 28% had BI-RADS density code 1; 40% code 2; 27% code 3; and 5% code 4 [7] (Table 1). In total, 416 breast cancers, including DCIS, were screen-detected, and there were 162 interval cancers. Screening sensitivity varied considerably across BI-RADS density code, from 78% in women with code 1 to 75% in those with code 2, 69% in those with codes and to only 47% in women with code 4. In the small group of women with extremely dense breasts; screening detected less than half of the breast cancers likely to have been present at the time of screening. Screening specificity was high at all levels of BI-RADS density.

### 3.2. Breast Cancer Incidence

In the Copenhagen pioneer program, the 61,741 women with fatty breasts accumulated 158,017 person years, during which 315 breast cancer cases were diagnosed (screen-detected, interval cancers, or cancer more the 2 years since last screen), and the 71,823 women with mixed/dense breasts accumulated 155,110 person years, during which 694 cases were diagnosed. Comparing the latter group with the first, gave an age-adjusted rate ratio of 2.45 (95% CI 2.14–2.81) [4] (Table 1).

The 54,808 women screened in the Capital Region in 2012/13 had in total, 578 breast cancers diagnosed within 2 years of their last screens (screen-detected and interval cancers). Compared with women with BI-RADS density code 1, women with successively higher density codes had an increased risk of breast cancer, with relative risks of 1.6 (95% CI 1.2–2.0), 1.9 (95% CI 1.4–2.3) and 2.0 (95% CI 1.3–2.8) for codes 2, 3 and 4, respectively [7] (Table 1).

### 3.3. Cohort Pattern

The density data from the two pioneer programs from Copenhagen and Funen showed the well-known decrease in density with increasing age; this was seen both in the cross-sectional data and by age within each birth cohort. We noted, however, that there was also variation across birth cohorts; for a given age, a larger proportion of women from younger birth cohorts had mixed/dense breasts [5]. As hormone users, on average, have higher densities than non-users [11], differences across birth cohorts in hormone use could potentially explain the cohort pattern. Individual data on prescribed and purchased drugs were available since 1992 for all women from Funen. Linkage with the hormone data from Funen confirmed that a higher proportion of hormone users than non-users had mixed/dense breasts, but the age and cohort patterns prevailed for both users and non-users. Therefore, differences in use of postmenopausal hormone therapy across birth cohorts cannot explain the cohort pattern of density.

### 3.4. Cross-Sectional Risk Factors

The Diet, Cancer and Health study took place in Copenhagen in 1993–1997, recruiting persons aged 50–64. About 5700 women from this study were screened in Copenhagen in 1993–2001. This allowed for cross-sectional studies of associations between risk factors reported in the health study and mammographic density registered at screening, and all estimates were adjusted for age and BMI, among other factors. The presence of mixed/dense breasts was statistically significantly more common among current hormone users compared with never-users, OR 1.24 (95% CI 1.14–1.35) [12] (Table 2). A follow-up part of this study showed adjusting for the presence of mixed/dense breasts only slightly reduced the association between current hormone use and breast cancer; hazard ratios (HRs) were 1.87 (95% CI 1.40–2.48) and 1.76 (95% CI 1.32–2.34) without and with adjustment for mammographic density, and mediation was estimated to be 10% (95% CI 4–22%). Diabetes was associated with a decreased risk of mixed/dense breasts, OR 0.61 (95% CI 0.40–0.92) with a notable variation across treatment groups [13]. Cigarette smoking showed a borderline statistically significantly negative association with risk of mixed/dense breasts, OR 0.86 (95% CI 0.75–0.99), driven particularly by women starting smoking early [14]. No statistically significant association was found between physical activity [15], traffic noise [16], alcohol consumption [17], air pollution [18] and mammographic density.

### 3.5. Childhood Constitution, Density and Breast Cancer Incidence

Using a longitudinal perspective study, we investigated the association between childhood body constitution and mammographic density at screening adjusting for age. Denmark, back in time had annual school health examinations, including measurements of weight and height from age 7 to age 13. Data for children from Copenhagen born 1930–1983 have been computerized. A previous follow-up study found that lean girls had a high risk of breast cancer in adulthood: relative risk 0.94 (95% CI 0.91–0.97) per 1-unit increase of BMI at age < 8 years, and 0.96 (95% CI 0.93–0.99) at age 8–14 years [19]. For the subgroup of 13,958 women who participated in the pioneer Copenhagen screening program, we found that a high proportion of lean girls had mixed/dense breasts at the first screen; the OR for mixed/dense breasts was 0.56 (9%% CI 0.53–0.58) per z-score of BMI at age 13. We also found that the excess breast cancer risk of lean girls was mediated by their mammographic density; the HR for breast cancer was 0.92 (0.84–1.00) per z-score of BMI at age 13 without control for density, and this estimate changed to a HR of 1.01 (95% CI 0.93–1.11) after control for density [10] (Figure 1).

## 4. Discussion

### 4.1. Main Results

At present, about one in three women in the screening age of 50–69 years had dense breasts (BI-RADS density codes 3 or 4). The Danish data confirmed the previously reported association of poorer screening sensitivity and greater breast cancer risk with greater mammographic density. Childhood body constitution has a strong association with breast tissue and breast cancer risk in mid-life age. Lean girls developed mixed/dense breasts, and they had an increased risk of breast cancer that was statistically explained by, and therefore, could be caused by, their mixed/dense breasts.

As reported previously, the proportion of women with mixed/dense breasts was lower for older women, but our data indicated that this decrease occurred at a later age in recently born cohorts compared with cohorts born longer back in time. The proportion of women with mixed dense breasts decreased with increasing age both for current and never users of hormone therapy. However, at a given age, more hormone users than non-users had mixed/dense breasts, indicating that the age and menopause-related decline in breast density on average occurred later in hormone users than in non-users. On this basis, both density and hormone use are expected to affect the risk of breast cancer at a given age. This was confirmed by our cohort data, as the risk of breast cancer was increased in hormone users when controlled for density; and increased in women with mixed/dense breasts when controlled for hormone use. Mammographic density was largely independent of other lifestyle factors in adults, bringing attention to the importance of the early-life exposures.

### 4.2. Results in Context of Literature

The 28% of women with BI-RADS density code 1 found in the Danish data is high from an international perspective. In the US Breast Cancer Surveillance Consortium data, only 9% of women screened had BI-RADS density code 1, but it should be taken into account that these data included women from the age of 35 years and above [20]. In both Norway [21] and Sweden [22], 16% of screened women had BI-RADS density code 1. However, density coding varies substantially between radiologists [7]. Differences between health care systems, e.g., the presence in the US of the Breast Density Notification Law [23], probably also play a role in explaining these differences.

From our data, we found that women with mixed/dense breasts are 2.5 times more likely to be diagnosed with breast cancer than women with fatty breasts. This corresponds rather well with the observed 5-year rate ratio of 2.20 and 2.41 for women with BI-RADS density codes 3 and 4 compared with women with BI-RADS density code 1 in the US Breast Cancer Surveillance Consortium data [20]. Our Danish findings also correspond well with the Boyd et al. case-control data from Canada, where the ORs were 1.8, 2.1 and 2.4 for women with densities 10–25%, 25–50% and 50–75%, respectively, compared to women with <10% density [3]. However, Boyd et al. found an OR of 4.7 in the 4% of women with density ≥75% [3]. Despite the fact that our BI-RADS density code 4 group constituted 5% of women, we did not find a similarly extreme breast cancer risk in this group.

Data linking measured body composition in childhood with mammographic density in mid-life age are rarely available. It is, therefore, very interesting that from studying 989 women similarly in the MRC 1946 British cohort, an OR of a greater Wolfe grade of 0.56 (95% CI 0.49–0.64) per 2.8 kg/m^3^ in BMI at age 15 was found [24]. This is similar to our OR of 0.56 (95% CI 0.53–058) per z-score of BMI at age 13. From similarly studying 490 women in the Tasmanian Longitudinal Health Study, negative regression coefficients were found for the association between childhood BMI and both absolute and percent mammographic density [25]. Similar results were found from studying 163 women from the New York Women’s Birth Cohort for weight changes from 1 to 7 years and absolute and percentage densities [26], and in 661 women from the Early Determinants of Mammographic Density study, in which childhood weight data were available up to the age of 4 [27]. Our study is, thus, clearly the largest one on this topic, and although we and the UK researchers used categories, whereas the Australian and US studies used continuous measures, it is reassuring that the results of the studies all point in the same direction. The fact that mammographic density, and therefore also risk of breast cancer in mid-life age, is to a large extent determined statistically by body composition in childhood, underlines the long-term time perspective needed in the primary prevention of breast cancer.

The likely decrease in mammographic density with increasing age has been proposed to reflect changes in the breast tissue related to menopause. Our finding of a later decrease in mammographic density across birth cohorts might, therefore, reflect changes in biological aging. However, the density codes in the Danish pioneer screening programs were given for pragmatic reasons only, and we could not exclude the possibility that the cohort pattern was derived from a change of coding practice. It was, therefore, interesting that a tendency for higher density in younger birth cohorts was observed also by a large Dutch study that used automated density coding [28]. The Danish and Dutch density data indicate that biological aging might come at a chronologically later age in recent, compared with earlier, generations. The fact that hormone users had on average, higher densities than non-users at a given age can be seen in the same perspective, as hormone therapy after menopause replaces the function of the natural hormones.

### 4.3. Strengths and Limitations

Our studies were all based on population-based data from organized screening programs, and the use of a comprehensive register data ensured full follow-up of all cohort members. In the risk factor analysis, we took advantage of the presence of a number of other databases, and it is a strength that these exposure data were all collected independently of the screening activity.

The risk factor analyses had, however, to be restricted to women present in the common part of screening and risk factor data. The Odense University Pharmaco-epidemiological Database covered all prescribed and purchased drugs from women from 1992 onwards, and the Funen screening program started in 1993, so hormone therapy data could be retrieved from all screened women. The Diet, Cancer and Health study recruited persons age 50–64 in 1993–1997, about one third of invited persons participated, and data were not available on all variables from all participants. Out of roughly 30,000 women screened at least once in Copenhagen at age 50–64 in 1993–1997, about 5700 could be included in the risk factor analysis (Table 2). The Copenhagen School Health Register included women born between 1930 and 1983 and attending school in the Copenhagen municipality. To be included in the childhood body constitution analysis, these women should have still been living in the Copenhagen municipality at screening age. Out of 134,640 women screened at least once in Copenhagen in 1991–2001, 13,958 could be included in the childhood body constitution analysis. The possibility of selection bias, therefore, has to be considered regarding the risk factor results.

The dichotomized code for mammographic density from 1991–2001 was used for administrative reasons only: to decide on whether a given woman should have one or two-view mammography at next screen. Fatty breasts were intended to correspond to BI-RADS code 1 and part of code 2 [4]. A later, small validation study showed that only two out of 120 (=1.7%) mammograms originally coded as fatty were recoded to BIRADS 3 or 4 [6]. The BI-RADS density codes used in 2012–2013 were coded independently by the two readers of the mammograms, and the highest code for a given mammogram was used as the consensus code. There was not full agreement between the two readers in the density coding. Reader 1 coded 34% of the mammograms with code 1; reader 2 coded 35% of the mammograms with code 1; but only 28% of the mammograms were given code 1 by both readers [7].

The fact that the risk of breast cancer is higher and the sensitivity of screening lower in women with dense breasts than in women with fatty breasts should be taken into account in the planning of personalized screening. Our data, furthermore, indicated that the proportion of women with dense breasts at a given age was higher in recently than in earlier born cohorts, a pattern expected to affect the need for resources in personalized screening.

## 5. Conclusions

In conclusion, the Danish data confirmed the patterns seen elsewhere, that screening sensitivity decreased with increasing mammographic density, and that women with dense breasts had a higher incidence breast cancer than women with fatty breasts. Furthermore, our Danish data showed that childhood body constitution had a strong association with the composition of breast tissue at middle-age. Breast density was lower for older women, and this aging process was associated both with hormone use and birth cohort. The mammographic density data accumulated in the Danish breast cancer screening program provided valuable insight into breast cancer aetiology.

## Figures and Tables

**Figure 1 jcm-08-02021-f001:**
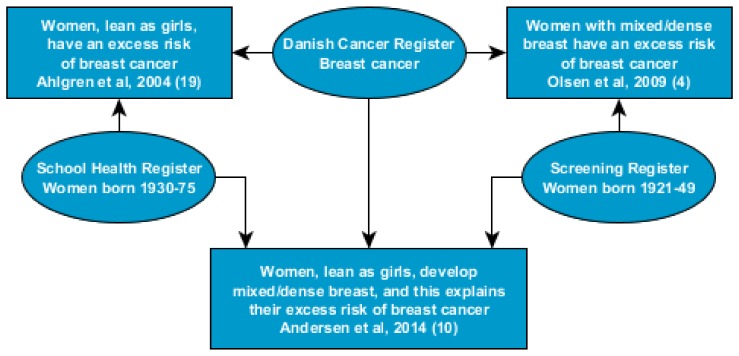
Danish studies on the association between childhood body constitution, mammographic density in mid-life age and risk of breast cancer. Boxes: studies; ovals: data sources.

**Table 1 jcm-08-02021-t001:** Women screened in Copenhagen’s municipality 1991–2001 by dichotomized density code, and women screened in Capital Region, Denmark, November 2012–December 2013 by BI-RADS density code.

Period/ Density (Reference)	Number of Women	% Women	Number of Breast Cancers ^1^				Sensitivity %	Specificity %	Rate Per 1000 ^5^/ % Women with Breast Cancer ^6^	Rate Ratio ^5^/ Relative Risk of Breast Cancer 95% CI ^6^
			Total	Screen- Detected	Interval Cancers	2+ Years from Last Screen				
1991–2001 [4] Total	134,640	100%	1009	471	279	259	63%	NA^4^	3.22	-
Fatty	61,741 ^2^	46%	315^2^	159^2^	65	91	71%	NA^4^	1.99	1
Mixed/ dense	71,823 ^2^	54%	694^2^	312^2^	214	168	59%	NA^4^	4.47	2.45 (2.14–2.81) ^7,8^
2012–2013 [7] Total	54,808	100%	578	416	162	NA^3^	72%	98%	1.05	-
BI-RADS 1	15,587	28%	113	88	25	NA^3^	78%	99%	0.72	1
BI-RADS 2	21,673	40%	245	184	61	NA^3^	75%	97%	1.13	1.6 (1.2–2.0) ^7^
BI-RADS 3	14,787	27%	184	127	57	NA^3^	69%	97%	1.24	1.9 (1.4–2.3) ^7^
BI-RADS 4	2761	5%	36	17	19	NA^3^	47%	98%	1.30	2.0 (1.3–2.8) ^7^

Note: ^1^ including ductal carcinoma in situ; ^2^ excluding cancers detected at first screening; ^3^ cohort followed for only 2 years after their last screen; ^4^ cannot be calculated, as the number of false positive screens were not tabulated; ^5^ rate per 1000 person years for 1991–2001; ^6^ proportion of screened women for 2012–2013; ^7^ age-adjusted; ^8^ person-years and cases accumulated based on last density category l.

**Table 2 jcm-08-02021-t002:** Odds ratios between exposures recorded in the Danish Diet Cancer and Health study 1993–1997 and presence of mixed/dense breasts at first participation in the Copenhagen breast cancer screening program 1993–2001.

Risk Factor (Reference)	N Women	Risk Factor Classification		Odds Ratio Exp/Non-Exp for Mixed/Dense ^2^	Comments and Selected Subgroup Results
		Exposed	Non Exposed		
Physical activity [15]	5703	+Do-it-yourself	−Do-it-yourself	1.06 (0.93–1.21)	Estimates for other activities closer to 1
Hormone therapy [12]	4501^1^	Current hormone use	Never hormone use	1.24 (1.14–1.35)	Breast cancer risk: Current/never: 1.87 (1.40–2.48) Adj. density: 1.76 (1.32–2.34); Density mediation: 10% (4–22%)
Traffic noise [16]	5260	Modelled traffic noise at address 5 years before mammogram		0.94 (0.86–1.03) per 10 dB	
Alcohol consumption [17]	5356	Alcohol consumers	No drinking	1.01 (0.81–1.27)	> 7 drinks/week at age 20–29: 1.31 (1.00–1.72)
Diabetes [13]	5644	+Diabetes	−Diabetes	0.61 (0.40–0.92)	Diet only: 0.56 (0.27–1.14) Oral drug: 0.59 (0.32–1.09) Insulin: 2.08 (0.68–6.35)
Cigarette smoking [14]	5356	Current smoker	Never smoker	0.86 (0.75–0.99)	Smoking <16 years: 0.79 (0.64–0.96) Current smokers, ≤5 pack-years: 0.62 (0.43–0.89)
Air pollution [18]	4769	Modelled nitrogen oxides at address 1970–1993/7		0.96 (0.93–1.01) per 20 μg/m^3^	

Notes: ^1^ postmenopausal women only; ^2^ adjusted for age and BMI, among other factors.

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
