# Peer review of "Mammographic Density and Screening Sensitivity, Breast Cancer Incidence and Associated Risk Factors in Danish Breast Cancer Screening"

_jcm, 2019, doi:10.3390/jcm8112021_

Round 1

Reviewer 1 Report

The authors have been (rapidly) very responsive to the reviewers' comments and I am very pleased. I noted a typo in the title of Supplementary Table 1.

Author Response

Reviewer 1:

Typo in Supplementary Table 1.

Answer: The typo has been corrected.

Reviewer 2 Report

This an interesting linkage study. I have a few comments which if considered in a minor revision will improve the quality of the paper.

Abstract

Please provide a clear explanation of "mixed/dense" in terms of BI-RADS

Line 25: recently born cohorts, how recent? provide years

line 29: "controlle" to "controlled" or "adjusted"

How was traffic noise determined?

Introduction: This section is very scanty. The authors need to provide a good overview of the subject, highlight the deficiencies in the literature that informed this study and provide a clear rationale for this study.

First sentence: what is the difference between invited screened women and screened women?

Materials and methods:

Mammograms with no signs of cancer were coded as fatty or mixed/dense breast... Fatty was equivalent to BI-RADS density code q and part of code 2. This statement is unclear. BI-RADS categories are clearly defined and the definitions used in BI-RADS should be used here so that the results can be easily understood. If the fatty code refers to BI-RADS 1&2 combined and mixed/dense breast tissue represent BI-RADs 3&4 combined, this needs to be clearly stated. Also, does this mean only images without cancer were used for this study? If so, how were sensitivity and specificity determined across density categories?

line 73: if the highest score was used where there are discrepancies, it cannot be a consensus. Consider a suitable term for this. Also, in how many cases did this discrepancy occur?

Calculations: for the first set of mammograms (1991-2001) fatty and mixed/dense: what was used as reference for calculating relative risk? I would guess it is the fatty breast but this needs to be clearly stated.

Also, given that only images with no cancer were coded, how were sensitivity and specificity calculated? See question above

Breast cancer incidence:

lines 65-67: Is this supposed to be incidence of breast cancer or risk of breast cancer?. Why report incidence for the pioneer program and cancer risk for the 2012/13 cohort?

Cohort pattern

line73: change "thah" to "than"

Cross sectional risk factors:

How were data on noise, air population and physical activity obtained?

Discussion: a summary of the findings has been provided and compared to the literature. However, it is unclear how these findings are relevant to policy and practice. A couple of statements highlighting the implications and relevance of these findings to policy and practice will do more justice to this paper.

Results in the context of the literature

line 53: "linking" nit "liking"

Conclusion

The last part of the conclusion is not supported by the results. the conclusions should also capture cancer incidence and diagnostic performance in relation to breast density.

Author Response

Reviewer 2:

Please provide a clear explanation of Mixed/dense in terms of BI-RADS

Answer: We have added:

”(fatty equal to BI-RADS 1 and part of 2; and other mixed/dense)”

Line 25: recently born cohorts, how recent? Provide years

Answer: It is a gradual development. In order to keep the text in the Abstract short we have chenged the text to:

”Percent of women with mixed/dense breast decreased with age, but at a higher level the later the women were born.”

Controlle

Answer: Sorry for the typo. It has now been corrected.

How was traffic noise determined?

Answer: This is explained in Table 2. In order to not make the Abstract too long, we have not added the description here.

Introduction: the section is very scanty.

Answer: We have followed the model of an Introduction with three short paragraphs. Our studies are discussed in light of the international literature in the Discussion. We hope this is acceptable.

Introduction: First sentence what it the difference between invited screened and screened women?

Answer: We have changed the sentence to:

”In population-based breast cancer screening programs all women in the target age-group are invited to screening, but not all of them attend. In these settings, breast cancer screening has been found to reduce breast cancer mortality by 20-25% for the entire population of invited women, and by 35% for women who attended and were actually screened.”

Material and Methods: First point: The BI-RADS definitions should be used.

ANSWER: We cannot change the codes used in Denmark in the beginning of the 1990s. At that time ”Fatty” was set to be equivalent to BI-RADS density code 1 and part of code 2.

Material and Mathods: Second point: Does this mean that only images without cancer were used in the study?

ANSWER: Sorry if this was not clear. It is explained in line 68-69: ”no density code was available for breast cancers detected at first screen”. We have expanded the text and added:

 ”but all breast cancers detected at subsequent screens had a density code.”

Line 73: First point: Highest score cannot be a consensus score.

Answer: The reviewer is right, we have stopped the sentence after highest score.

Line 73: Second point: discrepensies

Answer: These discrepensies are described on page 9, lines 104-106.

Calculations: First point: what was used as reference?

Answer: This is described on page 4, lines 19-20.

Calculations: Second point: only images with no cancer were coded …

Answer: See answer to question 8 above. See also line 83 describing the linkage with the Danish Cancer Register.

Breast cancer incidence: Why report incidence for the pioneer program and cancer risk for the 2012/13 cohort?

Answer: Good question. For the pioneer program we had long follow-up and person years had been counted. For women screened in 2012/13 we had only a two year follow-up, and person years had not been counted. This the explanation for the different calculation methods. This is described on page 4, lines 17-23.

Cohort pattern: Thah

Answer: Sorry for the typo, it has now been corrected.

Cross sectional risk factors: How were data on noise, air pollution and physical activity obtained?

Answer: This is described in Table 2.

Discussion: relevance for policy and practice

Answer: We have added:

 ”The fact that the risk of breast cancer is higher and the sensitivity of screening is lower in women with dense breasst than in women with fatty breasts should be taken into account in the planning of personalised screening. Our data furthermore indicated that the proportion of women with dense breast at a given age was higher in recently than in earlier born cohort;, a pattern expected to affect the need for ressources in personalised screening.”

Results in the context of the literature: Linking

Answer: We have corrected the typo.

Conclusion: First point: last part not supported by data.

Answer: We assume that the reviewer refers to the sentence: ”Breast density was lower for older women, and this aging process was associated both with hormone use and birth cohort”. We cannot follow the argument from the reviewer that this is not supported by the results. First, mixed/dense breast was more common in current hormone users than in never users (see page 5, lines 85-86). Second, at a given age more women in later born cohorts had mixed/dense breast than women born earlier, and this difference could not be explained by differences in hormone use (see page 5, lines 78-79).

Conclusion: Second point: Cancer incidence and diagnostic performance should be included.

Answer: We have added:

 ”In conclusion, the Danish data confirmed the patterns seen elsewhere that screening sensitivity decreased with increasing mammographic density, and that women with dense breasts had higher breast cancer incidence than women with fatty breasts. Furthermore…”

This manuscript is a resubmission of an earlier submission. The following is a list of the peer review reports and author responses from that submission.

Round 1

Reviewer 1 Report

attached

Reviewer 2 Report

This paper investigated the relationships between risk factors, mammographic density, and breast cancer risk using Danish Cancer Registry, screening register linked with other available data. Although the data are unique, there is little novelty in research questions, methods, and findings. There is not much additional information this study provides beyond what we already know from previous studies. I feel this study is no more than just confirming what has been published previously. Further, the paper needs extensive revision and clarification in interpretation of study findings. Specific comments are as follow:

Methods: I suggest to include a figure or table showing all cohort/registry data that were used/linked in this study and what variables/study design each data provide. Figure 1 is not informative at all. Figure 1 is no more than a list of data. Figure 1 needs to be clarified; is it missing an arrow? What is the point of showing figure 1? I suggest to modify if decide to keep. Does each woman in the study provide one screening data? Or can each woman provide multiple screenings? If so, which one did the authors choose to include? Did the authors examine the longitudinal change in mammographic density? Methods/Calculation: First setence (line 87-89) “defined as…” needs to be clarified. Does “+” (line 89) mean “and” or “positive”? Please clarify. Line 114-116: what is “first group” and what is “second group”. Please write out and clarify.      Table 1: Since mammographic density is strongly associated with both age and BMI. I suggest to adjust for age AND BMI, not age only as shown in table 1. Why are RRs from BIRADS crude, not age adjusted? Line 140: Unless the study longitudinally examined the change in mammographic density over time, I do not think it’s appropriate to say “delay” in age-related change in breast density. Line 141: does hormone use refer to postmenopausal hormone therapy? The birth cohort differences persisted in non-users; this is not consistent with author’s explanation “differences across birth cohorts in hormone use potentially explain the cohort pattern). Line 149: How was it a cross-sectional study? Were risk factor and mammographic density measured around the same time? Please clarify. Line 193: “unaffected by lifestyle factors in adult age”---mammographic density is also affected by adult BMI.